# Advances in the Genetic Basis and Molecular Mechanism of Lesion Mimic Formation in Rice

**DOI:** 10.3390/plants11162169

**Published:** 2022-08-21

**Authors:** Jiajie Yan, Yunxia Fang, Dawei Xue

**Affiliations:** College of Life and Environmental Sciences, Hangzhou Normal University, Hangzhou 311121, China

**Keywords:** rice, lesion mimic mutant, gene cloning, molecular mechanism, disease resistance

## Abstract

Plant lesion mutation usually refers to the phenomenon of cell death in green tissues before senescence in the absence of external stress, and such mutants also show enhanced resistance to some plant pathogens. The occurrence of lesion mimic mutants in rice is affected by gene mutation, reactive oxygen species accumulation, an uncontrolled programmed cell death system, and abiotic stress. At present, many lesion mimic mutants have been identified in rice, and some genes have been functionally analyzed. This study reviews the occurrence mechanism of lesion mimic mutants in rice. It analyzes the function of rice lesion mimic mutant genes to elucidate the molecular regulation pathways of rice lesion mimic mutants in regulating plant disease resistance.

## 1. Introduction

Plant lesion mutation refers to the phenomenon whereby plants spontaneously form necrotic spots of different sizes on the leaves, leaf sheaths, stems, and even the grains without external abiotic stress or biotic stress [1,2]. Related studies have shown that the generation process of plant lesion traits is very similar to the symptoms of the plant hypersensitivity response (HR) in vivo. The HR in plants is an innate immune mechanism for fighting against pathogen infection in vivo [3]. Plants further acquire systemic acquired immunity (SAR) by activating the HR process, thereby enhancing plant resistance. The HR is also a type of programmed cell death (PCD), and lesion mimic mutant plants generally exhibit the phenomenon of cell death, which results in premature senescence, consequently impacting the agronomic traits of some crops [4,5]. In addition, some studies have confirmed that most plant lesion mutants can enhance the resistance to plant pathogens. For example, in plants, reactive oxygen species (ROS) accumulation and the upregulation of defense genes of related pathogens usually occur in combination to enhance the resistance of plants to other diseases such as bacterial blight and blast fungus [6,7]. Therefore, such mutants are ideal materials for studying PCD and plant defense mechanisms.

In this paper, the origin, characteristics, and pathogenesis of rice lesion mimic mutants were reviewed. According to the biological functions of cloned rice lesion mimic genes, the molecular pathways involved in regulation were summarized. This is expected to have a certain impact on the study of rice disease resistance, molecular breeding, and improving rice yield and quality.

## 2. Discovery, Excavation, and Classification of Rice Lesion Mimic Mutants

Since the first lesion mimic mutant *sekiguchi lesion* (*SL*) in rice was reported in the mid-1960s [8], researchers have identified a series of other similar mutants. However, relatively few lesion mimic mutants have been produced under natural mutation, and most have been obtained by artificial mutation. The first cloned lesion mimic gene in rice was *Spotted Leaf 7* (*SPL7*), and the mutant *spl7*, which had small red-brown lesions distributed on the entire leaf surface, was induced by γ-ray irradiation [9]. The *SPL7* gene encodes a heat shock transcription factor, negatively regulating cell apoptosis under high-temperature stress after mutation. Subsequently, homologous genes of *SPL7* were found in maize, Arabidopsis, tomato, and other plants, which could produce lesion mimic traits in the corresponding plants [10]. In the lesion mimic mutants, gene mutation breaks the metabolic balance in the plant, leading to the appearance of abnormal phenotype and other damages, such as the lessening of the chlorophyll content, decrease of photosynthesis and plant height, and the final reduction of crop yield [11]. For example, the leaves of rice *lesion mimic and senescence mutant 1* (*lms1*) began to appear as russet brown spots at the late tillering stage, and the disease spots spread to the whole leaves and even stems with the plant growth. After the heading stage, the plant showed senescence symptoms, and the stems, leaves, and panicles were significantly dry and rapidly decayed, thereby affecting important traits such as 1000-grain weight and seed setting rate [12]. Due to the relative complexity of lesion mimic genes and the lack of in-depth functional analysis of lesion mimic genes in rice compared to model plants such as Arabidopsis, it is necessary to further study the function of lesion mimic genes in rice and continuously improve our understanding of the mechanism of the lesion mimic phenomenon [13].

Most of the lesions appear on the leaves or leaf sheaths in rice and are often brown, red-brown, dark brown, and dark yellow, among which brown lesions are the most common [14]. Lesions throughout the whole growth period, from seedling stage to mature seed stage leaves, always have obvious lesions. At the initial stage of vegetative growth, lesions appear between one month and two months after sowing. Some lesion mimic mutants continue to display the characteristics of lesions at the whole growth stage, and some lesions do not show any longer. Initiation lesions of reproductive growth occur only at the late stage of growth, and some occur after heading until seed maturation [15]. Therefore, lesion mimic mutants can be divided into whole life lesion mimics (WLLMs) (such as *lrd35*, *lrd40*), vegetative initiation lesion mimics (VILMs) (such as *lrd41*, *lrd44*), and reproductive initiation lesion mimics (RILMs) (such as *lrd27*, *lrd28*, *lrd39*) according to the occurrence period of the lesion mimic mutations. Another classification is to classify lesion mimic mutants into initiation class and propagation class according to their phenotypes [16]. In addition, lesion mimic mutants can also be divided into environmentally sensitive mutants and environmentally insensitive mutants according to the environmental sensitivity of the lesion mimic traits [14].

## 3. Mechanism of Rice Lesion Occurrence

The mechanism of lesion mimic occurrence in rice is extremely complex and is mainly regulated by disease resistance mechanisms, death regulation, and related metabolic enzymes (Figure 1). The changes in various signaling pathways in rice and the stress of some external environmental factors are also very important factors in forming a lesion.

### 3.1. Related Gene Mutation or Expression Change

In rice, if the disease-resistant or stress-resistant genes are mutated, it may cause disorder of the signaling pathways, resulting in PCD and further causing lesion mimic phenotype. For example, the rice *NLS1* gene encodes a coiled-coil nucleotide-binding leucine-rich repeat-like (CC-NB-LRR-like) R protein. After the mutation of *NLS1*, excessive H_2_O_2_ and salicylic acid (SA) accumulated in the mutant, and resistance-related genes were also upregulated, which led to the spontaneous formation of lesion mimic spots on the leaf sheath [17]. In addition, the phenotype of the dominant *Spl18* mutant was related to the insertion of the T-DNA activation label, which enhanced the transcriptional level of rice acyltransferase *OsAT1*, and caused lesion mimicry of *Spl18*. Meanwhile, the expression levels of resistance-related genes were upregulated with the severity of the lesion mimicry in *Spl18* [18]. In addition to rice, such phenomena have also been observed in other plants, such as with the *SSI* gene in Arabidopsis. The expression of defense genes in mutant Arabidopsis plants increased significantly, resulting in the lesion mimic phenomenon [19].

### 3.2. Accumulation of ROS

Reactive oxygen species are single-electron-reduction products of oxygen in vivo and mainly include hydroxyl radicals, hydrogen peroxide, and superoxide anion radicals. When the concentration of ROS in plants is lower than a certain value, the plant defense system is activated, and the ROS participates in the signal transduction of cells in plants. However, when the concentration of ROS exceeds a certain level, it will have toxic effects on cells and can even lead to cell death [20]. It has been found that plant lesion mutants typically produce a large amount of ROS during the formation of necrotic lesions. These high concentrations of ROS can destroy the normal structure of the cells and can also be used as signaling molecules to induce HRs, leading to cell death [20]. For example, the rice lesion mimic gene *SPL5* encodes a hypothetical transcription splicing factor 3b subunit 3 (SF3b3), which makes the mutant accumulate excessive superoxide anion and hydrogen peroxide, leading to the appearance of the lesion mimic phenotype in the mutant [21]. The same is true for the *nitric oxide excess1* (*noe1*) mutant in rice. The *NOE1* gene encodes a catalase OsCATC, which is responsible for eliminating hydrogen peroxide. So, the mutation of *NOE1* led to the production of excess hydrogen peroxide in leaves and the excessive accumulation of nitric oxide in the mutant because of the activation of nitrate reductase, which ultimately resulted in the exhibition of a lesion mimic phenotype in *noe1* [22].

### 3.3. Effects of Plant Hormones

In the process of plant lesion formation, some plant hormones, including ethylene (ET), jasmonic acid (JA), and salicylic acid (SA), play important roles in regulating different defense responses [23,24]. The *OsEDR1* gene in rice positively regulates the synthesis of ET, and the knockout (KO) of *OsEDR1* inhibits the ACC synthase genes, which encode the rate-limiting enzymes of ET biosynthesis resulting in the decreased production of ET. Meanwhile, the SA and JA contents in the *OsEDR1*-KO plants were increased, indicating that when the function of the *OsEDR1* gene was missing, the plant hormone level would be unbalanced, thus inducing necrotic lesions on the leaves at the booting stage [25]. Salicylic acid is one of the key hormones affecting plant resistance. In a study of the *OsSSI2* gene in rice, it was found that the transposon insertion mutants of this gene and RNA interference (RNAi) plants all showed lesion mimic traits, and the resistance of plants to *Magnaporthe grisea* was enhanced. Further studies showed that the endogenous SA content in transgenic plants was significantly increased [26]. In addition, when pathogens invade plants, the content of JA rapidly accumulates in plants and activates defense responses. As in lesion mimic mutants *oshpl3* and *spl29*, the JA content was significantly enhanced, and the expression levels of defense-related genes were upregulated [27,28].

### 3.4. Disorder of Plant Metabolic Pathways

In addition to being influenced by plant hormones, plant growth and metabolism are also regulated by many enzymes and proteins. If these enzymes and proteins are inactivated in the process of metabolism, they will cause metabolic disorders during plant growth, resulting in the emergence of lesion mimic phenotypes. The *FGL* gene of rice encodes a protochlorophyllide oxidoreductase B (OsPORB), which catalyzes the photoreduction of protochlorophyllide to chlorophyllide under high light conditions. After the mutation, the chlorophyll metabolism of *fgl* mutant was disordered, and the leaf of *fgl* in color was rapidly changed from green to yellow and then formed lesions during leaf elongation in field-grown conditions [29]. The *SPL28* gene encodes a grid-associated receptor protein complex, namely the central subunit μ1 (AP1M1), which participates in the vesicle transport process of the Golgi body. The mutation of *SPL28* disordered the normal material transport of the Golgi body, and the red-brown spots on leaves of *spl28* mutant began to emerge at the early tillering stage and reach the maximum number at the heading stage [30].

### 3.5. Uncontrolled PCD

Programmed cell death is a common death mode in the development of organisms and is determined by genes. In the plant growth process, once the normal PCD in the body is disordered, it may lead to abnormal growth and development [31]. The appearance of the lesion mimic phenotype in plants is also one of the manifestations of spontaneous cell death. In rice, the *SPL11* gene encodes an E3 ubiquitin ligase, negatively regulating PCD. The mutation of this gene led to a change in E3 ubiquitin ligase activity, resulting in the generation of lesion mimic spots [32]. In addition, in rice, the *Oryza sativa accelerated cell death and resistance 1* (*OsACDR1*) gene and *G-box factor 14-3-3 homologs* (*GF14e*) gene are also related to PCD reaction. *OsACDR1*, also named *OsEDR1**,* encodes a putative Raf-like mitogen-activated protein kinase kinase kinase (MAPKKK), which can inhibit OsMPKK10.2 activity through physical interaction. The mutation of *OsACDR1* released the inhibition of OsMPKK10.2, so the activation of the pathogen-inducible OsMPKK10.2-OsMPK6 cascade was amplified [33]. Meanwhile, the mutation promoted plant cell death, resulting in the lesion mimic phenotype, and the defense-related genes were upregulated [25,33]. *GF14e*, encoding a 14-3-3 protein, is induced during effector-triggered immunity (ETI) associated with pathogens. *GF14e* has a negative regulatory effect on cell death and the defense response of rice. Therefore, its RNAi plants exhibited the lesion mimic character as well as increased resistance to bacterial blight and other pathogens [34].

### 3.6. Influence of the External Environment

When a plant is subjected to abiotic environmental stresses, such as light, temperature, or humidity, it will elicit a series of immune stress responses to environmental stress. This process is often accompanied by the accumulation of ROS and the initiation of PCD process, resulting in the appearance of a lesion mimic phenotype. The rice *lesions stimulating disease resistance 1* (*OsLSD1*) gene can be induced under light conditions and inhibited under dark conditions. Moreover, *OsLSD1* plays a negative regulatory role in PCD, and so, the *lsd1* mutant can produce lesion mimic trait under light [35]. In addition, the mutant *lesion mimic and premature senescence 1* (*lmps1*) are also affected by light. When part of the leaf was shaded, the lesion mimic phenotype did not appear on the mutant leaves, but following the reintroduction of light, the lesion mimic character appeared again on the leaves [36]. A study on six *lrd* lesion mimic mutants in rice found that high temperature could inhibit the lesion mimic trait of mutants *lrd31*, *lrd35,* and *lrd40*, while low temperature could significantly promote the occurrence of lesion mimic traits (except for the *lrd31*), and at a normal temperature of 28 °C, all *lrd* mutants produced lesion mimic traits on the leaves [37]. Therefore, environmental factors such as light intensity and temperature also affect plant lesion mimic production.

## 4. Identification, Cloning, and Functional Analysis of Rice Lesion Mimic Genes

With the development of molecular biology, many studies have been conducted on rice lesion mimic mutant materials. More than 70 lesion mimic mutants have been identified in rice, and some have been successfully cloned (Table 1). According to the genetic characteristics of the identified rice lesion mimic mutants, most of them are controlled by single recessive genes [1]. Function analysis of the cloned lesion mimic genes indicated that these genes are mainly involved in lipid metabolism, gene transcription regulation, chlorophyll metabolism, and plant defense responses (Figure 2).

### 4.1. Plant Lipid Metabolism Pathway

Fatty acids and their derivatives play important signal transduction roles in plant defense responses. The rice *OsSSI2* gene encodes a fatty acid dehydrogenase, which is responsible for the dehydrogenation of stearic acid(18:0) to oleic acid(18:1). Knockdown (kd) of the *OsSSI2* gene reduces the oleic acid level, increases the stearic acid level in plants, and the *OsSSI2*-kd plants showed a lesion phenotype, increased endogenous free SA content, and significant upregulation of resistance-related genes such as *WRKY45*. Compared with the wild type, the resistance of *OsSSI2*-kd plants to *Magnaporthe grisea* and *Xanthomonas oryzae* pv. *oryzae* was also significantly enhanced, indicating that *OsSSI2* could reversely regulate the defense stress response of rice [26]. In addition, the rice *OsPLDβ1* gene encodes a phospholipase D, including an HKD motif and a C2 domain. The RNAi plants of this gene exhibited a lesion mimic phenotype on the leaves without the interference of external pathogens, and the accumulation of ROS and some plant antitoxins also occurred in the plants at the late development stage. In addition, RNAi plants spontaneously activated plant defense responses and significantly enhanced resistance to other pathogens such as rice blast [41].

The *OsHPL3* gene encodes a hydroperoxide lyase, whose insertion mutation leads to the functional inactivation of the product, resulting in the appearance of the lesion mimic phenotype in the mutant. The JA and SA contents in the mutant plant were significantly higher than those in the wild type, and thus, the defense signaling pathway was activated. Compared with wild-type plants, mutant plants showed enhanced resistance to bacterial blight, while *OsHPL3* overexpression transgenic plants showed enhanced resistance to the brown rice planthopper [27].

### 4.2. Gene Transcription Regulation Pathway

In rice, *SPL7* was the first cloned lesion mimic gene that encodes a heat stress transcription factor. A single base mutation in the *SPL7* gene, namely a change in an amino acid in the binding domain from tryptophan to cysteine, resulted in the change of the target protein domain, which resulted in the loss of function of the target protein and ultimately the lesion mimic phenotype [9,10]. In addition, the rice *SPL5* gene can encode a splicing factor subunit 3b, which negatively regulates the defense stress response and cell death mainly by splicing the mRNA precursors of genes related to plant cell death or immune responses [45].

The *OsLMS* gene encodes a negative regulator related to the stress response in rice, which has a carboxyl-terminal domain (CTD). The domain of CTD is related to RNA polymerase II (RNAP II), and RNAP II is the core component of the transcription complex, which can catalyze mRNA synthesis. Thus, it can participate in various mRNA maturation processes or gene transcription regulation. The RNAi lines showed a lesion mimic phenotype when *OsLMS* gene expression was inhibited, which confirmed that the *OsLMS* gene mutation was the cause of the lesion mimic phenotype. *OsLMS* is homologous to the *FIERY2/CPL1* gene in *A. thaliana*, which controls many plant growth processes, such as the stress response and growth and development. The *lms* mutant showed sensitivity to low-temperature stress at the early growth stage, indicating that *OsLMS* is related to the rice stress response [47].

### 4.3. Chlorophyll Metabolism Pathway

The *RLIN1* gene encodes a porphyrinogen III oxidase, which is involved in the tetrapyrrole biosynthetic pathway. Tetrapyrrole pigments are the largest and most widely distributed pigments in nature. The molecular structure of tetrapyrrole derivatives is contained in chlorophyll. The mutant *rlin1* can produce a lesion mimic phenotype [44], which indicates that the metabolic pathway of chlorophyll synthesis is also involved in the formation of rice lesions. The *FGL* gene must maintain chlorophyll light-dependent synthesis during leaf development, especially under strong light irradiation. Under constant light or strong light irradiation, the new green leaves of the *fgl* mutant rapidly turn yellow and form lesion mimic spots. Further studies have found that the *FGL* gene encodes OsPORB, which is necessary for chlorophyll synthesis and metabolism [29]. The rice lesion mimic gene *ELL1* is also related to chloroplast development. In the mutant *ell1*, the contents of photosynthetic pigments such as chlorophyll a, chlorophyll b, and carotenoids were significantly decreased, the endocytic structure of the chloroplast was degraded, and the photosynthetic protein activity was significantly weakened [42].

### 4.4. Plant Defense Response Pathway

*OsLSD1* encodes a protein with three conserved zinc-finger domains, which has an antagonistic effect on the PCD process. Studies have demonstrated that the antisense inhibition of *OsLSD1* could produce a lesion mimic phenotype in transgenic plants, and the expression of defense-related genes was upregulated, indicating that *OsLSD1* may negatively regulate the defense response of rice [35].

*Oryza sativa* Pto-interacting protein 1a (OsPti1a), a plasma membrane protein kinase-related protein, acts as a negative regulator of innate immunity and signal transduction defense in rice. Therefore, the *OsPti1a*-overexpression transgenic plants were more susceptible to bacterial blight and other pathogens than wild-type plants [62]. When the expression of the *RAR1* gene was inhibited, the resistance of the mutant was also inhibited, which indicated that *OsPti1a* might negatively regulate the *RAR1*-mediated defense mechanism of rice disease resistance [39].

The *rlr1* mutant showed premature senescence and a lesion mimic phenotype under natural growth conditions, which was accompanied by the large accumulation of ROS, and the *rlr1* mutant also enhanced the resistance to rice blast and bacterial blight. Studies have shown that the transcription factor OsWRKY19 can activate the immune response of *OsPR10*, and the *OsRLR1* gene-encoding protein can interact with the transcription factor OsWRKY19. Therefore, the rice defense mechanism associated with *OsRLR1* may be mediated by the transcription factor OsWRKY19 [61].

In addition to the above regulatory pathways, lesion mimic genes in rice are also involved in regulating inter-cellular transport pathways and ROS systems in plants. For example, *SPL28*, mentioned above, encodes an AP1M1 protein located in the Golgi apparatus, and the AP1M1 protein participates in the vesicle transport process of the Golgi apparatus [30]. The *SPL5* and *NOE1* genes are related to the regulation of the ROS signaling pathway [21,22].

## 5. Disease Resistance of Rice Lesion Mimic Mutants

Currently, according to the reported results, most rice lesion mimic mutants show enhanced resistance to pathogens. For example, compared with the wild type, the *lsd1* mutant significantly improved the resistance to bacterial and fungal pathogens [35]. The rice lesion mimic mutant *spl28* significantly increased resistance to bacterial blight and rice blast due to the accumulation of large amounts of callose in the plants [30]. In addition, *spl10*, *cdr3*, and other mutants showed enhanced resistance to rice blast; *spl21*, *lmes1*, *hm83*, and other mutants exhibited enhanced resistance to bacterial blight, and the mutant *lmm1* was resistant to both sheath blight and blast [1]. These studies showed that the lesion mimic mutant gene might activate the immune stress response in plants due to the loss of function, thereby enhancing the resistance to pathogens. Therefore, it also shows that the lesion mimic mutant of rice is a good material for studying plant PCD procedures and disease resistance and defense mechanisms.

### 5.1. Lesion Mimic Genes Involved in Plant Hormone Regulation

While many rice lesion mimic mutants enhance disease resistance, the content of plant hormones in them usually changes, among which salicylic acid (SA) is a key hormone affecting the acquired resistance of plant systems. When plants are subjected to some biological stress, some plant tissues will have an allergic reaction, which leads to the accumulation of SA content, activating some PR protein expression and ultimately enhancing the disease resistance of plants. Jasmonic acid (JA) is also an important plant hormone. When pathogens invade plants, JA accumulates rapidly in plants and activates defense stress response pathways to prevent pathogen invasion [63]. The content of SA and JA in many lesion mimic mutants is often changed when the disease resistance is enhanced. *NahG* gene is the key gene for the decomposition of SA, and the phenotype of lesion mimics in some mutants is inhibited to varying degrees by hybridization with its transgenic plants and various lesion mimic mutants, indicating that SA plays a crucial role in the formation of lesion mimics [64]. Salicylic acid content was significantly increased in the *HM47* mutants of rice, and the resistance of *HM47* to bacterial blight was significantly increased. The experimental results showed that the increase of resistance of *HM47* to bacterial blight was achieved by regulating the metabolic pathway of salicylic acid [65]. Another example is *OsNPR1* in rice. *OsNPR1* is a key regulator in the SA-mediated disease resistance pathway, and *OsNPR1* may be involved in the antagonistic interaction between the SA and JA signaling pathways in rice [38]. Recent research found that *OsNPR1* upregulated the expression of the indole-3-acetic acid (IAA)-amide synthase gene (*OsGH3.8*) by interfering with the auxin signaling pathway, ultimately affecting the normal growth and development of rice [66]. Silencing of *OsEDR*, a lesion mimic gene in rice, resulted in significantly increased contents of SA and JA in the RNAi plants, thereby initiating defensive responses [25].

### 5.2. Lesion Mimic Genes Involved in ROS Signaling Pathway

For plants, ROS has strong biochemical activity, which can cause oxidative damage to some intracellular functional substances such as proteins, DNA, and lipids. Therefore, a series of relatively perfect antioxidant mechanisms have been developed in plant cells to protect cells from ROS invasion, such as the system for scavenging ROS by enzymatic reactions such as superoxide dismutase (SOD) and catalase (CAT) [67]. Studies have shown that the accumulation of ROS near the lesion can be detected in most rice lesion mimic mutants, which indicates that ROS may play a very important signal transduction role in forming a lesion. For instance, *OsAPX2*, encoding an ascorbate peroxidase, can scavenge ROS to protect seedlings from abiotic stress. While the loss function of OsAPX2 leads to decreased ascorbic acid peroxidase activity and the excessive accumulation of hydrogen peroxide and malondialdehyde in cells, and finally, in the mutant *Osapx2**,* the rice leaves show lesion mimic traits [68]. A study of the rice lesion mimic gene *GF14e* found that when the expression of the *GF14e* gene was interfered with by RNAi technology, lesion mimic spots of different sizes were produced in the leaves of the transgenic plants, and ROS accumulated in the lesion, and the resistance to related pathogens was enhanced [34]. However, some lesion mimic genes may be directly involved in the resistance mechanism of plants through the metabolism of peroxides. For example, the rice lesion mimic gene *NOE1* can encode a catalase. In the *noe1* mutant, excessive H_2_O_2_ is produced in its leaves due to the loss of the gene function, resulting in the appearance of the lesion mimic phenotype in the *noe1* mutant, but the resistance of the mutant is also enhanced [22].

In addition, some studies have found that there is not only ROS accumulation in many rice lesion mimic mutants but also endogenous SA levels in mutant plants will change. Subsequent studies confirmed a certain interaction between SA and ROS, and ROS was involved in SA-induced signal transduction pathways. On the one hand, SA can activate the activity of ROS-producing enzymes (such as NOX) to promote the accumulation of intracellular ROS. On the other hand, SA slows down the ROS scavenging by inhibiting the activity of ROS scavenging enzymes (such as CAT and APX) to a certain extent [69,70]. Further studies have found that catalase is one of the targets of SA, suggesting that SA may inhibit intracellular ROS accumulation by inhibiting CAT activity [71]. In the CAT-deficient transgenic plants, the H_2_O_2_ content was increaseed with the increase of light intensity, and the production of SA was stimulated, resulting in lesion mimic mutants and leaf wilting [72]. For example, the content of defense hormone SA in the *ebr1* mutant was significantly increased, and the content of hydrogen peroxide in the plant was also significantly increased, which improved the disease resistance of the mutant [51]. The *nls1* mutant accumulates excessive hydrogen peroxide and SA, and the upregulated expression of defense-related genes enhances the resistance to bacterial diseases [17].

## 6. Research Prospect

As a major food crop, rice is important in ensuring global food security. Therefore, it is particularly important to address the issues of rice diseases and insect pests. There are many factors affecting the generation of rice lesion mimic mutants. In addition, a certain relationship exists between the generation of lesion mimic mutants and increased pathogen resistance in rice. Most of the lesion mimic mutants improve the resistance to rice blast and bacterial blight. Still, at the same time, they also affect the agronomic traits of rice, such as the plant height, panicle length, grain number per panicle, and seed setting rate because of PCD; thus, striking a balance between disease resistance and PCD is still an urgent issue to be considered. The cloning and functional study of lesion mimic genes in rice have become particularly relevant. However, the molecular mechanisms associated with rice lesions are not fully understood, and thus, greater in-depth research should be carried out in the future

In addition, as rice is a model plant for genomics research, exploring the mechanism of cell death and disease resistance in rice plants is a popular research avenue. Since the generation of plant lesions can lead to the upregulation of defense gene expression and the enhancement of plant resistance to pathogens, lesion mimic mutants in rice are also ideal materials for studying the mechanism of PCD and disease resistance. This research also plays an important role in rice breeding and production practice. Therefore, cloning and functional analysis of rice lesion mimic genes can further elucidate the resistance mechanism of rice and provide a more powerful theoretical basis for the breeding of highly resistant rice.

## Figures and Tables

**Figure 1 plants-11-02169-f001:**
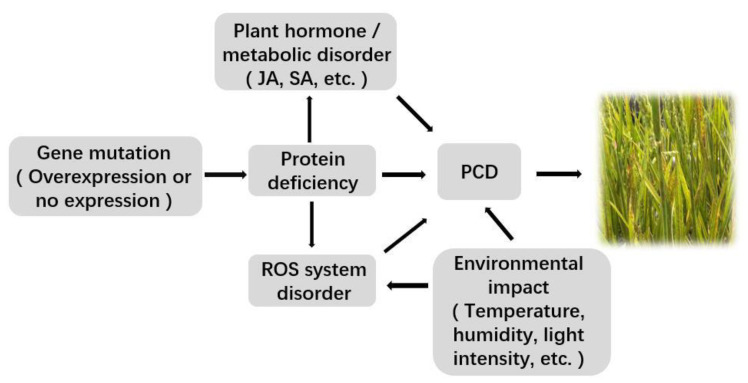
Mechanism of rice lesion mimic occurrence. (The mechanism of plant lesion formation is varied and complex and mainly regulated by genes such as disease resistance, death regulation, and basic metabolic enzymes; plant defense signaling molecules and external environmental factors also play important roles in forming lesions. All the arrows indicated positive regulatory relations).

**Figure 2 plants-11-02169-f002:**
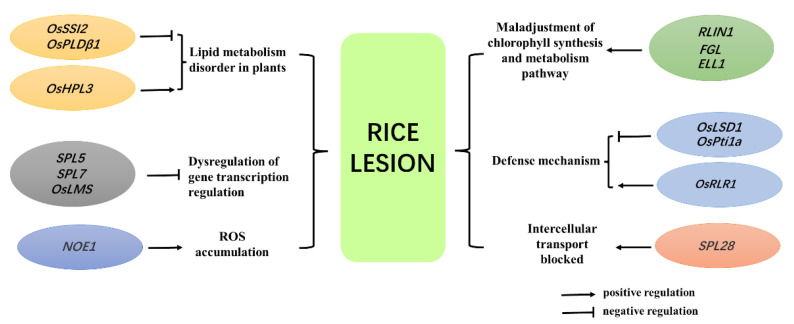
Part of the regulation pathway of lesion mimic genes in rice. (The regulatory pathways involved in rice lesion mimic genes mainly include lipid metabolism and chlorophyll synthesis metabolism in plants, transcription pathways of plant cell death, defense pathways of plants, intercellular transport pathways, and ROS signal transduction pathways).

**Table 1 plants-11-02169-t001:** Partially cloned 35 rice lesion mimic genes.

Mutant Gene	Mutant Type	Accession Number	Protein Function Analysis	Reference
*SPL7*	RILM	Os05g05304	Heat shock transcription factor	[9]
*SPL11*	RILM	Os12g05700	E3 ubiquitin ligase	[32]
*LSD1*	VILM	Os08g01595	C2C2 zinc finger protein	[35]
*OsNPR1(NH1)*	VILM	Os01g01943	Transcriptional coactivators	[38]
*SPL18(OsAT1)*	WLLM	Os10g01956	Acyltransferase	[18]
*OsPti1a*	VILM	Os05g01358	Rice protein kinase	[39]
*XB15*	VILM	Os03g60650	Protein phosphatase	[40]
*OsACDR1* *(OsEDR1,SPL3)*	RILM	Os03g01601	Mitogen-activated protein kinase	[33]
*OsSSI2*	RILM	Os01g09199	Fatty acid dehydrogenase	[26]
*OsPLDβ1*	VILM	Os10g38060	Phospholipase	[41]
*SPL28*	RILM	Os01g07036	Subunits of grid-related receptor protein complexes	[30]
*OsSL(ELL1)*	RILM	Os12g02680	Cytochrome P450 monooxygenase	[42,43]
*RLIN1*	WLLM	Os04g06108	Porphyrin III oxidase	[44]
*GF14e*	WLLM	Os02g05803	14-3-3 protein	[34]
*NOE1*	RILM	Os03g03910	Catalase	[35]
*NLS1*	RILM	Os11g14380	CC-NB-LRR type R protein	[17]
*SPL5*	RILM	Os07g02037	Splicing factor 3b subunit	[45]
*OsHPL3*	VILM	Os02g01102	Hydroperoxide lyase	[28,46]
*OsLMS*	WLLM	Os02g06390	RNA binding protein	[47]
*CslF6*	VILM	Os08g01605	Cellulose-like synthase F	[48]
*RLS1*	VILM	Os02g10900	NB-ARM domain protein	[49]
*FGL*	VILM	Os10g35370	OsPORB protein, involved in cytochrome synthesis	[29]
*SPL29*	WLLM	Os08g02069	Acetylglucosamine pyrophosphatase	[27]
*LMR*	RILM	Os06g01300	Adenosine triphosphatase	[50]
*EBR1*	VILM	Os05g19970	E3 ubiquitin ligase in RING domain	[51]
*OsPLS1*	RILM	Os06g45120	Vacuolar proton ATPase subunit	[52]
*SPL32*	VILM	Os07g06584	Ferroxin-dependent glutamate synthase	[53]
*SPL33*	WLLM	Os01g01166	eEF1A-like protein	[54]
*OsCUL3a*	VILM	Os02g07460	Cullin protein	[55]
*LML1*	WLLM	Os04g06599	Eukaryotic Release Factor 1 Protein	[56]
*SDS*	VILM	Os01g57480	SD-1 receptor-like kinase	[57]
*SPL35*	WLLM	Os03g02050	CUE domain protein	[58]
*SPL40*	WLLM	Os05g03120	Ribosome structural components	[59]
*LMM24*	VILM	Os03g24930	receptor-like cytoplasmic kinase	[60]
*OsRLR1*	RILM	Os10g07978	CC-NB-LRR protein	[61]

The lesion mimic mutants can be divided into whole life lesion mimics (WLLMs), vegetative initiation lesion mimics (VILMs), and reproductive initiation lesion mimics (RILMs) according to the occurrence period of the lesion mimic mutations.

## Data Availability

Not applicable.

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
