# Peer review of "Advances in the Genetic Basis and Molecular Mechanism of Lesion Mimic Formation in Rice"

_plants, 2022, doi:10.3390/plants11162169_

Round 1
Reviewer 1 Report
The review MS by Yan et al., presents an interesting idea to cover the mechanisms of lesion mic formation in rice. Unfortunately, the ‘molecular mechanisms’ are scattered and not conclusive. Just enlisting the genes does not pass as a ‘mechanism’. The MS is also lacking in scientific and English writing. Based on my review of the MS, I suggest significant changes and reorganizations in the MS. Please find below my specific comments
Please mention clearly the direction of ‘change (positive or negative, imbalance (up or down) in the relevant places. It is hard to associate with the phenotype and whether the mutation in a certain gene leads to what kind of phenomenon.
The gene products and their functions are not described. Expand all the gene/mutant names at first mention. For example, sl, spl7, lms1, ird35,ird40, NLS1, CC-NB-LRR, ssi, SF3b3 etc. Please thoroughly check the MS for all other acronyms and expand those. Define scientific names for all the plant species at first mention
Most of the time authors have cited reviews instead of original works. Please check and correct throughout the MS. A few examples are L18-20: cite some original references. The same comment goes for L29-31. The reference cited for L2023 is not apt. Please cite a more relevant work. Cite reference for L47-49. Need reference for Lines 93-97
It would be helpful for the readers to better understand the context if the authors could kindly enlist the traits or define each of the whole life lesion mimics (WLLMs), vegetative initiation lesion mimics (VILMs), reproductive initiation lesion mimics (RILMs), initiation class and propagation class lmms
L49: ‘gene mutation also has a certain harm to the lesion mimic mutant’ reads strange to me. Are the LMMs not the result of gene mutation?
L83-87: ‘In addition, the phenotype of the rice spl18 mutant is related to the insertion of the T-DNA activation label. The insertion of the T-DNA activation label has a significant enhancement effect on the expression level of genes around the insertion site, resulting in a significant increase in the expression level of related disease-resistance genes in the mutant spl18, which is due to the change in gene expression and the occurrence of lesion mimic spots’. This reads confusing. Does it mean that rather than the mutation in the spl18 gene, it is the presence of the TDNA activation label in the disease-resistant genes? Also, there is a problem with a significant increase in the expression level of related disease-resistance genes in the mutant spl18, which is due to the change in gene expression and the occurrence of lesion mimic spots. The change in any gene’s expression is due to a change in its expression. How can the gene expression change be due to the occurrence of lesion mimic spots?
L113-115: please explain the gene function of EDR1 and not only the ultimate outcome. Also, explain how the imbalance in ethylene here would lead to necrotic lesions. What balance, is the ethylene down or up here? At what stage does it cause necrosis?
I recommend authors categorize all the described mutants as per WLLMS, VILMS, RILMS, initiation, propagation or environmentally responsive mutants.
L120: both SA and JA provide plant resistance in response to a specific group of pathogens which is bio/hemibiotrophic (SA) and necrotrophic (JA). Also, most often these two hormones have antagonistic functions, suppressing each other. So generalizing the positive role of JA in pathogen defense is fundamentally wrong here.
The purpose of L121-123 is not clear. Please reframe the sentence with a clear message.
L129-130: OsPORB catalyzes the reduction of protochlorophyllide to chlorophyllase: Do you mean it makes the chlorophyllase ENZYME?
130-132: please expand on how the disruption in fgl would only lead to necrotic spots and not the albino phenotype (in the absence of chlorophylls)
L132-137: please explain here how the disruption in vesicle transport would lead to lesion mimic phenotype. Why does it appear only after anthesis
L145: what is the gene product of ACDR1? Mention the enzyme/protein produced and specify how it regulates PCD
The same comment applies to GF14e. what is the specific function of 4-3-3 protein? How does it negatively regulate cell death?
What do you mean by ‘opening of PCD procedures’?
These are only a few examples of lapses in writing. I suggest applying my comments throughout the MS. Most often Authors just left the sentences in middle without further explanation
Figure 1 is connecting arrows everywhere. It fails to show the positive or negative relationship with PCD.
Figure 2 is too crowded and hard to understand. Please include a descriptive legend. How the negative regulation of defense genes leads to PCD. In the text, they mention otherwise! What do you mean by abnormal defense pathway? In the absence of 'direction' for any change or regulation mentioned in this figure, it is very hard to understand and the Figure appears obsolete
Authors should include how PCD /lesion mimic can affect crop yield. How PCD can be both a curse or a desirable train for plant yield under field conditions. Including more studies, please expand on this subtitle.
I recommend authors include an in-depth review of phytohormones and PCD to underline their relationship. Why did you not consider hormones other than SA, JA and ethylene? Are they not involved in the PCD? The ABA is an important senescing hormone, AUX is related to both plant growth and disease susceptibility. Likewise, CK has positive functions in both growth and disease resistance
Author Response
Please mention clearly the direction of ‘change (positive or negative, imbalance (up or down) in the relevant places. It is hard to associate with the phenotype and whether the mutation in a certain gene leads to what kind of phenomenon.
Response:Thank you for your advice. We have mentioned clearly the direction of change in some relevant places.
The gene products and their functions are not described. Expand all the gene/mutant names at first mention. For example, sl, spl7, lms1, ird35,ird40, NLS1, CC-NB-LRR, ssi, SF3b3 etc. Please thoroughly check the MS for all other acronyms and expand those. Define scientific names for all the plant species at first mention
Response:Thank you for your comments. We have added expanded names of gene /mutant, and the gene products and their functions in some parts of the MS.
Most of the time authors have cited reviews instead of original works. Please check and correct throughout the MS. A few examples are L18-20: cite some original references. The same comment goes for L29-31. The reference cited for L20-23 is not apt. Please cite a more relevant work. Cite reference for L47-49. Need reference for Lines 93-97
Response:Thank you for your advice. We have cited new references.
It would be helpful for the readers to better understand the context if the authors could kindly enlist the traits or define each of the whole life lesion mimics (WLLMs), vegetative initiation lesion mimics (VILMs), reproductive initiation lesion mimics (RILMs), initiation class and propagation class lmms
Response:Thank you for your suggestion. We have added the definition in the MS.
L49: ‘gene mutation also has a certain harm to the lesion mimic mutant’ reads strange to me. Are the LMMs not the result of gene mutation?
Response:Thank you. We have modified the sentence. Yes, LMMs are the result of gene mutation.
L83-87: ‘In addition, the phenotype of the rice spl18 mutant is related to the insertion of the T-DNA activation label. The insertion of the T-DNA activation label has a significant enhancement effect on the expression level of genes around the insertion site, resulting in a significant increase in the expression level of related disease-resistance genes in the mutant spl18, which is due to the change in gene expression and the occurrence of lesion mimic spots’. This reads confusing. Does it mean that rather than the mutation in the spl18 gene, it is the presence of the TDNA activation label in the disease-resistant genes? Also, there is a problem with a significant increase in the expression level of related disease-resistance genes in the mutant spl18, which is due to the change in gene expression and the occurrence of lesion mimic spots. The change in any gene’s expression is due to a change in its expression. How can the gene expression change be due to the occurrence of lesion mimic spots?
Response:Thank you. The sentences have been modified in the MS.
L113-115: please explain the gene function of EDR1 and not only the ultimate outcome. Also, explain how the imbalance in ethylene here would lead to necrotic lesions. What balance, is the ethylene down or up here? At what stage does it cause necrosis?
Response:Thank you for your advice. We have changed this sentence as follow. The OsEDR1 gene in rice positively regulates the synthesis of ET, and the knockout (KO) of OsEDR1 inhibited the ACC synthase genes, which encodes the rate-limiting enzymes of ET biosynthesis, resulting in the decreased production of ET. Meanwhile, the SA and JA contents in the OsEDR1-KO plants were increased, indicating that when the function of the OsEDR1 gene is missing, the plant hormone level will be unbalanced, thus inducing necrotic lesions on the leaves at the booting stage.
I recommend authors categorize all the described mutants as per WLLMS, VILMS, RILMS, initiation, propagation or environmentally responsive mutants.
Response:Thank you for your advice. We have categorize all the described genes in table 1.
L120: both SA and JA provide plant resistance in response to a specific group of pathogens which is bio/hemibiotrophic (SA) and necrotrophic (JA). Also, most often these two hormones have antagonistic functions, suppressing each other. So generalizing the positive role of JA in pathogen defense is fundamentally wrong here.
Response:Thank you. Most studies have confirmed that JA and SA are antagonistic hormones, but other studies found that JA and SA may have independent pathways. In the OsEDR1-KO plants, Shen et al(2011) found SA and JA contents were increased simultaneously. In the spl29, JA content was enhanced significantly, however, SA content was not mentioned, which is much hinger than JA content in plants. In the oshpl3 rice lesion mimic mutant, the level of JA was raised and bacterial blight resistance was enhanced. So, we generalized the role of JA in rice defense responses.
The purpose of L121-123 is not clear. Please reframe the sentence with a clear message.
Response:Thank you. This sentence was reframed as follow. Such as in lesion mimic mutants oshpl3 and spl29, the JA content was significantly enhanced, and the expression level of defense-related genes was upregulated.
L129-130: OsPORB catalyzes the reduction of protochlorophyllide to chlorophyllase: Do you mean it makes the chlorophyllase ENZYME?
Response:Thank you very much. We made an error and have replaced “chlorophyllase” by “chlorophyllide”.
130-132: please expand on how the disruption in fgl would only lead to necrotic spots and not the albino phenotype (in the absence of chlorophylls)
Response:Protochlorophyllide oxidoreductase (POR) catalyzes photoreduction of protochlorophyllide (Pchlide) to chlorophyllide in chlorophyll (Chl) synthesis, and is required for prolamellar body (PLB) formation in etioplasts. In which, OsPORB is essential for maintaining light-dependent Chl synthesis throughout leaf development. After mutation, the leaf of fgl in color was rapidly changed from green to yellow and then formed lesions during leaf elongation in field-grown condition. There may be phenotypic differences between different alleles of one gene. In fact, mutation of OsPORB does not mean complete loss of function.
L132-137: please explain here how the disruption in vesicle transport would lead to lesion mimic phenotype. Why does it appear only after anthesis
Response:Thank you. Spl28 encodes the subunit μ1 (μAP1M1) of the reticulin-associated receptor protein complex, which is located in the Golgi, and may be involved in regulating vesicle trafficking. Dysfunction of spl28 leads to HR-like lesions in plants, which leads to leaf senescence. The mutation of SPL28 disordered the normal material transport of the Golgi body, and the red-brown spots on leaves of spl18 mutant began to emerge at early tillering stage, and reach the maximum number at heading stage. However, why the mutant phenotype appeared or aggravated at this stage was not mentioned in the reference.
L145: what is the gene product of ACDR1? Mention the enzyme/protein produced and specify how it regulates PCD
Response:Thank you. The rice Oryza sativa accelerated cell death and resistance 1 (OsACDR1), also named OsEDR1, encodes a putative Raf-like mitogen-activated protein kinase kinase kinase (MAPKKK), which can inhibit OsMPKK10.2 activity through physical interaction. The mutation of OsACDR1 release the inhibition of OsMPKK10.2, so that the activation of pathogen-inducible OsMPKK10.2-OsMPK6 cascade is amplified[33]. Meanwhile, the mutation promotes plant cell death, resulting in the lesion mimic phenotype, and the defense-related genes are upregulated
The same comment applies to GF14e. what is the specific function of 4-3-3 protein? How does it negatively regulate cell death?
Response:Thank you. GF14e, encoding a 14 - 3 - 3 protein, is induced during effector‐triggered immunity (ETI) associated with pathogens.GF14e has a negative regulatory effect on cell death and the defense response of rice, therefore, its RNAi plants exhibit the lesion mimic character as well as increased resistance to bacterial blight and other pathogens.
What do you mean by ‘opening of PCD procedures’?
Response:Thank you for reminding. These words were improper, and have been instead by “initiation of PCD process”.
These are only a few examples of lapses in writing. I suggest applying my comments throughout the MS. Most often Authors just left the sentences in middle without further explanation
Response:Thank you for your advice. We have introduced the function of genes more carefully in the modified MS.
Figure 1 is connecting arrows everywhere. It fails to show the positive or negative relationship with PCD.
Response:Thank you for reminding. All the arrows indicated positive regulatory relations.
Figure 2 is too crowded and hard to understand. Please include a descriptive legend. How the negative regulation of defense genes leads to PCD. In the text, they mention otherwise! What do you mean by abnormal defense pathway? In the absence of 'direction' for any change or regulation mentioned in this figure, it is very hard to understand and the Figure appears obsolete
Response:Thank you for reminding and your suggestion. We have modified Figure 2.
Authors should include how PCD /lesion mimic can affect crop yield. How PCD can be both a curse or a desirable train for plant yield under field conditions. Including more studies, please expand on this subtitle.
Response:Thank you for your suggestion. How PCD /lesion mimic can affect crop yield was described in the part1. At present, PCD occurring rice lesion mimic mutants is basically unfavorable to the formation of crop yield and quality. this issue was discussed in part 6.
I recommend authors include an in-depth review of phytohormones and PCD to underline their relationship. Why did you not consider hormones other than SA, JA and ethylene? Are they not involved in the PCD? The ABA is an important senescing hormone, AUX is related to both plant growth and disease susceptibility. Likewise, CK has positive functions in both growth and disease resistance
Response:Thank you for suggestion. When writing this part, we noticed that a variety of hormones can promote plant growth and improve resistance. After carefully reading the references, we thought that some hormones were not highly correlated with the formation of lesion mimic, therefore, only SA, JA and ET in the lesion mimic mutants were described.
Reviewer 2 Report
The manuscript presents current knowledge about the phenomena of lesion mimic formation in rice plants.
The manuscript is well prepared.
It is properly sectioned, however the first section is missing it's title (Introduction).
The manuscript should be carefully checked and edited for small mistakes, such as punctuation. Especially the list of reference section.
Author Response
The manuscript presents current knowledge about the phenomena of lesion mimic formation in rice plants.
The manuscript is well prepared.
It is properly sectioned, however the first section is missing it's title (Introduction).
Response:Thank you. We have modified the manuscript.
The manuscript should be carefully checked and edited for small mistakes, such as punctuation. Especially the list of reference section.
Response:Thank you for your advice. We have checked again.
Reviewer 3 Report
Mutants that exhibit lesion mimic phenotypes are very easily found in nature because of their appearance. Each mutation is caused by a mutation in a gene that does not appear to be related to each other. Their phenotypes are diverse. These things make it difficult to even classify the mutants. The authors summarize rice lesion mimic mutants well in this manuscript.
This manuscript is excellent.
If the authors can include the picture of each mutant in Table1 will be the best.
Line 134 : disordered thenor-mal material à disordered the normal material
Author Response
Mutants that exhibit lesion mimic phenotypes are very easily found in nature because of their appearance. Each mutation is caused by a mutation in a gene that does not appear to be related to each other. Their phenotypes are diverse. These things make it difficult to even classify the mutants. The authors summarize rice lesion mimic mutants well in this manuscript.
This manuscript is excellent.
If the authors can include the picture of each mutant in Table1 will be the best.
Response:Thank you for your advice. The phenotypes of these lesion mimic mutants are similar. Most of the lesions appear on the leaves or leaf sheaths and are often brown, red-brown, dark brown, and dark yellow, among which brown lesions are the most common.
Line 134 : disordered thenor-mal material à disordered the normal material
Response:Thank you for reminding. We have corrected this sentence.